# Interactions between Cationic Dye Toluidine Blue and Fibrous Clay Minerals

Qingfeng Wu [1,*], Kristen Carlson [2], Qi Cheng [1], Xisen Wang [3] and Zhaohui Li [2,*]

1   School of Physics and Optoelectronic Engineering, Yangtze University, 1 Nanhuan Road, Jingzhou 434023, China; Cheng1139948254@163.com
2   Department of Geosciences, University of Wisconsin—Parkside, 900 Wood Road, Kenosha, WI 53144, USA; carls066@rangers.uwp.edu
3   Department of Chemistry, California State University, Sacramento, CA 95819, USA; x.wang@csus.edu
*   Correspondence: cjdxwqfscience@163.com (Q.W.); li@uwp.edu (Z.L.)

**Abstract:** Interactions between cationic dyes and negatively charged mineral surfaces have long attracted great attention from clay mineralogists, environmental scientists, and chemical engineers. In this study, the interactions between a cationic dye toluidine blue (TB) and palygorskite and sepiolite were investigated under different experimental conditions. The results showed that in addition to cation exchange, the specific surface area (SSA) of the minerals, particularly the formation of dimer molecules on the surface of both minerals, also accounted for the much higher TB uptake in comparison to their cation exchange capacities (CEC). The TB molecules were sorbed to the external surfaces, as no d-spacing expansion was observed in X-ray diffraction analyses. FTIR analyses showed strong interactions between the C=N or N-(CH$_3$)$_2$ group and the mineral surfaces, suggesting net electrostatic interactions if either of these functional groups bears a positive charge. Results from molecular dynamic simulations suggested dense monolayer TB formation on palygorskite because of its limited SSA and large CEC values. In comparison, a loosely dimeric formation was revealed on sepiolite for its large SSA and limited CEC values. Therefore, palygorskite is a better carrier for the sorption of cationic dyes, as evidenced by Maya blue paintings.

**Keywords:** dimers; interactions; mechanism; palygorskite; sepiolite; toluidine blue

## 1. Introduction

Earth materials are important resources. Among them, clay minerals, due to their large specific surface area (SSA), high cation exchange capacity (CEC), and bulk quantities, are of great potential in practical applications in modern society. As such, they have been studied extensively as sorbents for sorptive removal of different types of contaminants from water. Among the contaminants assessed, those of a cationic nature could be effectively removed by clay minerals via cation exchange processes.

Water-soluble dyes could be cationic, zwitterionic, or anionic in nature. Currently more than 10,000 types of dyes are used in different industries [1]. As different dyes have different physico-chemical properties, active approaches to remove dyes from solution include physical [2], chemical [1], and biological processes [3], and could be achieved via either sorption or degradation.

On the other hand, the use of clay materials as carriers for color dyes in painting could be dated back to more than a thousand years ago during Maya civilization, when Maya blue, a mixture of palygorskite with indigo dyes, was used for sculpture, artwork, and textiles in the Aztec area of Mexico, suggesting that ancient civilization had already utilized the strong sorption of color dyes by clay minerals. Among the common clay minerals, montmorillonite (MMT) is the most extensively studied, due to its large SSA and high CEC values.

Palygorskite (PAL) and sepiolite (SEP) are fibrous phyllosilicates. They have relatively large SSAs with internal channels parallel to [001] direction and are accessible by inorganic cations. As such, for the sorption and cation exchange processes, both the surface sites and the channel area are available for small cations and molecules via cation exchanges and specific sorption. As such, PAL and SEP have also been studied extensively for contaminant removal. Sorption of methylene blue (MB) and crystal violet (CV) on an Oman PAL resulted in a sorption capacity of 51 and 58 mg/g, corresponding to 159 and 140 mmol/kg, while the measured specific surface area (SSA) and cation exchange capacity (CEC) of the PAL were 92 m$^2$/g and 192 meq/kg, respectively [4]. In a different study, the sorption of MB reached a capacity of 48 mg/g, or 150 mmol/kg, on an untreated palygorskite with a CEC value of 230 mmol/kg [5]. Sorption of safranin O (SO), also a cationic dye, on an Iraqi PAL resulted in a capacity of 200 mg/g, or 570 mmol/kg, but the CEC value of the PAL was not reported [6]. The MB sorption capacity was 105 mg/g or 328 mmol/kg on a raw PAL, and the sorption capacity increased to 207 mg/g or 649 mmol/kg after mild hydrothermal modification using chloroacetic acid [7]. The sorption of MB was 78, 98, and 159 mg/g, or 244, 306, and 497 mmol/kg on three palygorskite samples, but the CEC values of the samples were not provided [8]. More recently, the sorption of SO on PAL was 195 mmol/kg [9] in comparison to its CEC value of 175 meq/kg [10].

On SEP, MB sorption reached a capacity of 80 mg/g, or 250 mmol/kg, and increased with decreasing solution pH, with the mechanism of MB sorption being attributed to interaction through a reaction between the hydroxyl end groups of SEP and the cationic group of MB [11]. SO sorption on iron oxide/sepiolite magnetite composite (MSep) reached a capacity of 18.48 mg/g or 68 mmol/kg [12], in comparison to 45 mmol/kg on a raw SEP [9], whose CEC value is 15 meq/kg [13].

Toluidine blue (TB) is a phenothiazine basic dye commonly used in biology and biotechnology to stain tissues rich in DNA and RNA [14]. It is also commonly used in the textile industry, medicine, and biotechnology as a mediator for various reactions [15]. Although its structure is similar to that of MB, studies on TB removal from water using Earth materials is limited.

Fuller's earth is any clay material capable of decolorizing oil or other liquids without harsh chemical treatment and its major component is palygorskite. TB removal by Fuller's earth obtained from Turnkey showed a capacity of 66 mmol/kg, and the sorption capacity increased to 400 mmol/kg after alkali-treatment [16]. Ion exchange was attributed to the major mechanism, but the cation exchange capacity of the material was not provided in the study. In another study, TB sorption on Turkish zeolite reached a capacity of 55 mg/g or 210 mmol/kg at pH 11 [15]. A capacity of 28 mg/g or 90 mmol/kg was found for TB sorption on gypsum [17]. In a different study, a bentonite showed a TB sorption capacity of 18.4 mg/g, or 70 mmol/kg in comparison to 2835 mg/g for malachite green [18]. On the contrary, TB sorption on a Na-montmorillonite (MMT), a major component of bentonite, was 5.8 mmol/g, or 1773 mg/g [19], but neither the CEC value of the mineral nor the mechanism of TB sorption on MMT was mentioned. Bentonite modified by graphene oxide increased TB uptake capacity to 458 mg/g at pH 8, in comparison to raw bentonite owing to the synergistic effect between bentonite and graphene oxide [20]. Unfortunately, the TB sorption on the raw bentonite was not provided in their study.

The purpose of this study was to investigate the mechanism of basic dye sorption by fibrous clay minerals under different physical and chemical conditions using TB as an example. To achieve this goal, extensive instrumental characterization was deployed to link the changes in material properties with the amount of TB sorption. To confirm the predicted surface configuration of sorbed TB molecules and to further illustrate the mechanism of TB removal by fibrous clay minerals, molecular dynamic simulation was performed.

## 2. Materials and Methods

### 2.1. Materials

The TB used is in a chloride form. It has a CAS number of 92-31-9, a molecular weight of 305.82 g/mol, a solubility of 30 g/L(Acros) (Fair Lawn, NJ, USA), and two pKa values at 2.4 and 11.6 (https://www.chemicalbook.com/ChemicalProductProperty_EN_CB66745 75.htm, accessed on 30 April 2021; [21]). With a planner molecule, it has a dimension of a 1.2 nm long by 0.52 nm wide (Figure 1).

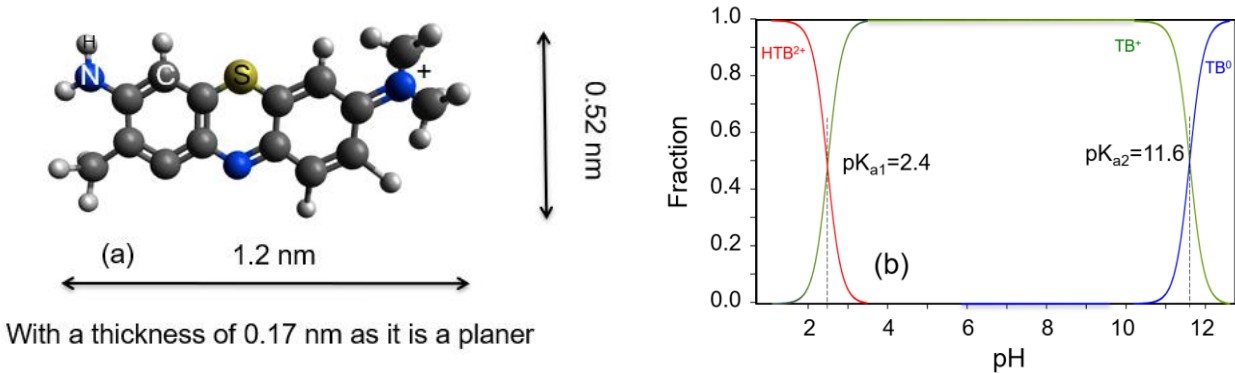

**Figure 1.** Molecular structure of TB (**a**) and its pH-speciation relation (**b**).

The PAL used is PFl-1 and the SEP used is SepSp-1. Both are standard clay minerals obtained from the Source Clay Minerals Repository. Their CEC values are 175 [10] or 195 meq/kg (https://www.clays.org/sourceclays, accessed on 30 April 2021) and 15meq/kg [13], and SSA values are 173 [22] or 136 $m^2$/g (https://www.clays.org/sourceclays, accessed on 30 April 2021) and 250 $m^2$/g [13] for PFl-1 and SpeSp-1, respectively.

### 2.2. Interactions between TB and Minerals in Solution

To evaluate the interactions between TB and clay minerals under different physico-chemical conditions, batch tests were conducted. In each experiment, 10 mL of TB solution and 0.2 g of PAL or 0.5 g of SEP were put into each 50-mL centrifuge tube. The initial concentration varied from 0.0 to 6.0 mM for the isotherm study and was at 5.0 mM for all other studies. The mixtures were shaken at 150 rpm and room temperature for 24 h, except for the kinetic study, in which the shaking varied from 0.1 to 24 h. The influence of solution pH on dye/mineral interactions was evaluated with the equilibrium solution pH values between 3 and 11 with an interval close to 1. The influence of solution ionic strength on dye/mineral interaction was evaluated by setting the solution ionic strength to 0.001, 0.01, 0.1, and 1.0 M of NaCl. The temperature studies were performed at 23, 33, 43, and 53 °C. At the end of the mixing period, samples were centrifuged at 3500 rpm for 10 min to separate the minerals from the solution and the supernatants were analyzed for equilibrium TB concentrations using a UV-Vis method after the supernatant was passed through 0.45 μm syringe filters. The amount of sorbed TB was calculated by the difference between the initial and equilibrium TB concentrations multiplied by the liquid/solid ratio. Duplicated experiments were conducted under each physico-chemical condition.

### 2.3. Instrumental Analyses

The equilibrium TB and counterion $Cl^-$ concentrations were determined by a UV-Vis spectrophotometer with a wavelength of 625 nm [17] and by an ion chromatography (IC) with a PRP-100 anion exchange column made by Hamilton (Reno, NV, USA). The X-ray diffraction (XRD) analyses were performed using a Shimadzu 6100 X-ray Diffractometer (Long Beach, CA, USA) for both minerals after their interactions with different initial TB concentrations. A Ni filtered CuKα radiation at 30 kV and 40 mA and a scanning speed of 2°/min were used for the experimental conditions. Samples were scanned from 5–40° ($2\theta$).

The FTIR spectra were obtained from a Shimadzu IRAffinity-1S FTIR spectrometer (Long Beach, CA, USA) utilizing a quartz attenuated total reflection device. Samples were scanned from 400 to 4000 cm$^{-1}$ with a resolution of 2 cm$^{-1}$.

### 2.4. Molecular Dynamic Simulation

The FORCITE module in Materials Studio 6.0 software (San Diego, CA, USA) was used for molecular dynamic simulation to investigate the interactions between the mineral surfaces and TB molecules, to optimize the geometry of TB molecules and the clay minerals, and to determine the surface configurations of sorbed TB on mineral surfaces at 298 K. The simulation models were built based on the crystal data of clay minerals and TB molecules. For PAL and SEP, their unit cell parameters were: $a$ = 12.78 Å, $b$ = 17.86 Å, $c$ = 5.24 Å, $\beta$ = 95.78°, Z = 4; and $a$ = 13.43 Å, $b$ = 26.88 Å, $c$ = 5.281 Å, Z = 4, respectively. The supercells were made of $2b \times 4c$ for both minerals. The number of TB molecules used per supercell was determined from the TB sorption capacity divided by the SSA of the minerals using 173 [22] and 250 m$^2$/g [13] for PFL and SEP, respectively, and then multiplied by the supercell size. They were 7 and 3 for PAL and SEP, respectively. After generating the initial coordinates of the supercells, the NVE ensemble (298 K, 1 × 10$^5$ Pa) simulations were used to reach equilibrium for 200 ps with 1 fs as the time step and all of 200 ps were applied to record data every 1 ps for analysis. The constructed model was optimized geometrically.

### 3. Results and Discussion

#### 3.1. Isotherms Study of TB Interactions with Both Minerals

Sorption of TB on both minerals was first evaluated as a function of initial TB concentrations, and thus, equilibrium solution concentration, which is the isotherm study (Figure 2a). Results were fitted better to the Langmuir model that has the form of:

$$C_s = \frac{K_L S_m C_L}{1 + K_L C_L} \tag{1}$$

and be converted into a linear form so that the parameters can be calculated linearly:

$$\frac{C_L}{C_s} = \frac{1}{K_L S_m} + \frac{C_L}{S_m} \tag{2}$$

where $C_L$ and $C_S$ are the equilibrium TB concentration in solution (mmol/L) and amount of TB sorbed on minerals (mmol/kg). The parameters $S_m$ (mmol/kg) and $K_L$ (L/mmol) are the TB sorption capacity on and affinity for the minerals. The fitted results are $S_m$ = 287 and 120 mmol/kg and $K_L$ = 658 and 522 L/mmol for TB sorption on PAL and SEP, respectively. In comparison, the $S_m$ values were 195 and 45 mmol/kg and $K_L$ values were 2450 and 77 L/mmol for SO sorption on the same PAL and SPE, respectively [9]. Although the TB is in a cationic form in this study, the $S_m$ values are much larger than the CEC values of 175 [10] and 15 [13] meq/kg for PAL and SEP. Sorption of SO on MSep also followed the Langmuir isotherm with a capacity of 18.48 mg/g or 68 mmol/kg [12]. A capacity of 200 mmol/kg was found for MB sorption on a PAL [23]. SO sorption on PAL at a capacity of 570 mmol/kg was also reported [6].

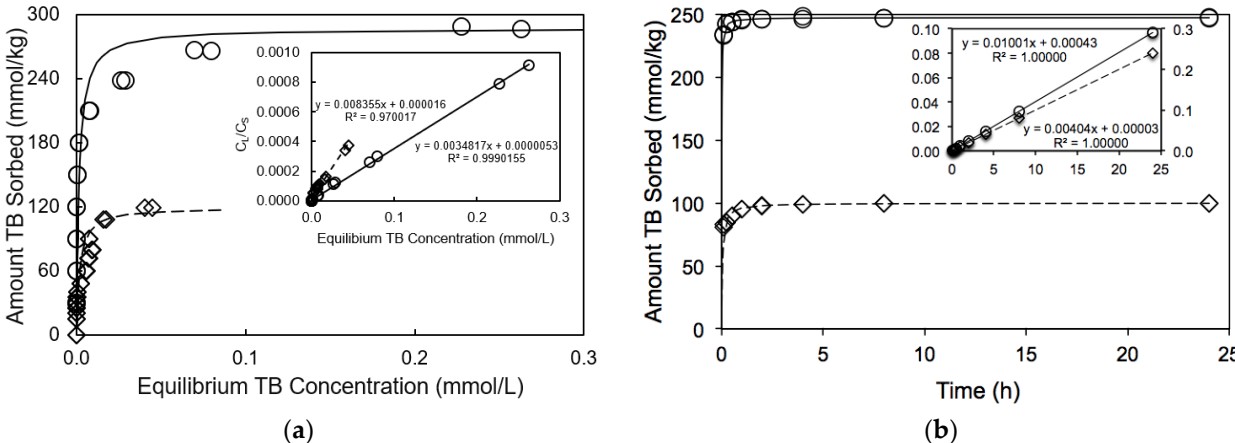

**Figure 2.** Isotherm (**a**) and kinetic study of TB sorption on PAL and SEP. The lines are Langmuir (**a**) and pseudo-second order (**b**) fits to the experimental data. The inset is the fit to their linearized form.

### 3.2. Kinetics of TB Interaction with Fibrous Clay Minerals

Figure 2b showed the sorption of TB on both minerals as a function of time. Generally speaking, TB sorption was fast and could reach equilibrium in about 2 h. The data were fitted to different kinetic models with the pseudo-second order kinetics achieving the best fit. It has the form:

$$q_t = \frac{kq_e^2\, t}{1 + kq_e t} \tag{3}$$

which can be converted into a linear form:

$$\frac{t}{q_t} = \frac{1}{kq_e^2} + \frac{1}{q_e}t \tag{4}$$

so that the kinetic parameters can be solved in a linear regression. Similarly, pseudo-second order kinetics were used to fit the SO sorption on MSep [12] and SO sorption on PAL and SEP [9]. In Equations (3) and (4), $k$ (kg/mmol-h) and $kq_e^2$ (mmol/kg-h) are the rate constant and initial rate of TB sorption on the minerals; and $q_t$ and $q_e$ (mmol/kg) are the amount of TB sorbed at time $t$ and at equilibrium. The fitted results are $q_e$ = 248 and 100 mmol/kg; $k$ = 0.6 and 0.3 kg/mmol-h; $kq_e^2$ = 35,714 and 2326 mmol/kg-h for TB sorption on PAL and SEP, respectively. These results suggest that the fibrous clay minerals, particularly PAL, is a good candidate for the sorption of cationic color dyes, which explains why PAL was used as a carrier in Maya blue painting.

As the TB sorption capacity was much larger than the CEC values of the minerals, in addition to cation exchange, other mechanisms may also contribute to the removal of TB from water. To assess whether the uptake of TB on mineral surfaces was in monolayer or dimeric configurations, equilibrium counterion $Cl^-$ concentrations were determined by an IC method. The results showed some uptake of $Cl^-$ accompanying TB sorption, with up to 10% and 50% $Cl^-$ sorption on PFL and SEP, respectively (Figure 3). This would account for significantly higher TB sorption in comparison to SO on the same minerals and to the CEC values of the minerals.

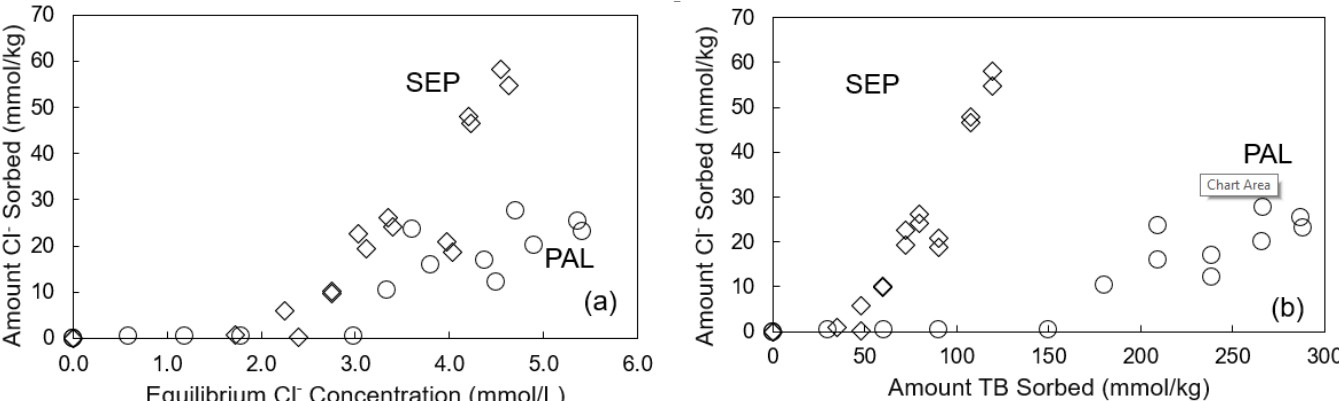

**Figure 3.** Uptake of counterion Cl⁻ accompanying TB sorption on PAL and SEP (**a**) and relationship between counterion Cl⁻ sorbed against TB sorbed (**b**).

### 3.3. Effects of Equilibrium Solution pH, Ionic Strength and Temperature on TB Sorption

Both the equilibrium solution pH and ionic strength had essentially no influence on TB sorption on both minerals (Figure 4a,b). A similar pH and ionic strength effect on SO sorption on the same minerals was observed earlier [9]. A previous result showed that the pH effect on SO removal by PAL was also minimal [6]. These results suggest a much higher affinity of cation dyes in comparison to inorganic cations for the mineral surfaces. In contrast, the sorption of MB on sepiolite was higher at acid pH [11]. However, under the acidic pH condition, the competition between H⁺ and MB⁺ should have a reduction in MB uptake. On the contrary, sorption of MB on palygorskite was higher at pH > 10, which is attributed to better dispersion in suspension and thus increased SSA [4].

Influence of temperature on TB sorption showed opposite trends between PAL and SEP (Figure 4c). The thermodynamic parameters of TB sorption are related to the solute distribution coefficient $K_d$, by

$$\ln K_d = -\frac{\Delta H}{RT} + \frac{\Delta S}{R} \tag{5}$$

where the $\Delta H$ and $\Delta S$ are the changes in enthalpy and entropy due to TB sorption, $R$ is the gas constant, and $T$ is the absolute temperature in $K$. The $\Delta H$ and $\Delta S$ values are related to the free energy $\Delta G$ of TB sorption by

$$\Delta G = \Delta H - T\Delta S \tag{6}$$

The thermodynamic parameters of TB sorption calculated from Equations (5) and (6) are –18 to –27 kJ/mol for $\Delta G$ (Table 1), indicating net attractive interactions between TB and mineral surfaces via physical sorption or electrostatic interactions. The $\Delta G$ value was about –15 kJ/mol for SO uptake on MSep, suggesting physi-sorption [12], and was –18 to –24 kJ/mol for SO sorption on both minerals [9]. In contrast to the $\Delta G$, the $\Delta H$ and $\Delta S$ values were positive for TB sorption on SEP, but negative for TB sorption on PAL. Previously, $\Delta H$ values were –14 to –16 kJ/mol and $\Delta S$ values were 0.01 to 0.03 kJ/mol for SO sorption on both minerals. The opposite trend of $\Delta H$ and $\Delta S$ values for TB sorption on SEP in comparison to SO sorption on the same SEP may suggest a change in sorption mechanism. This is most likely due to the extremely large TB sorption at 100 mmol/kg in this study in comparison to SO sorption at 20 mmol/kg in a previous study [9]. As the CEC of the SEP is only 15 meq/kg, the much larger TB sorption is likely due to aggregation of TB molecules on SEP surfaces bridged by counterion Cl⁻. Analyses of equilibrium counterion Cl⁻ by IC showed significant Cl⁻ removal from solution by SEP, suggesting dimeric or admicelle sorption of TB (Figure 3), in contrast to the dominant cation exchange mechanism for SO sorption on SEP [9].

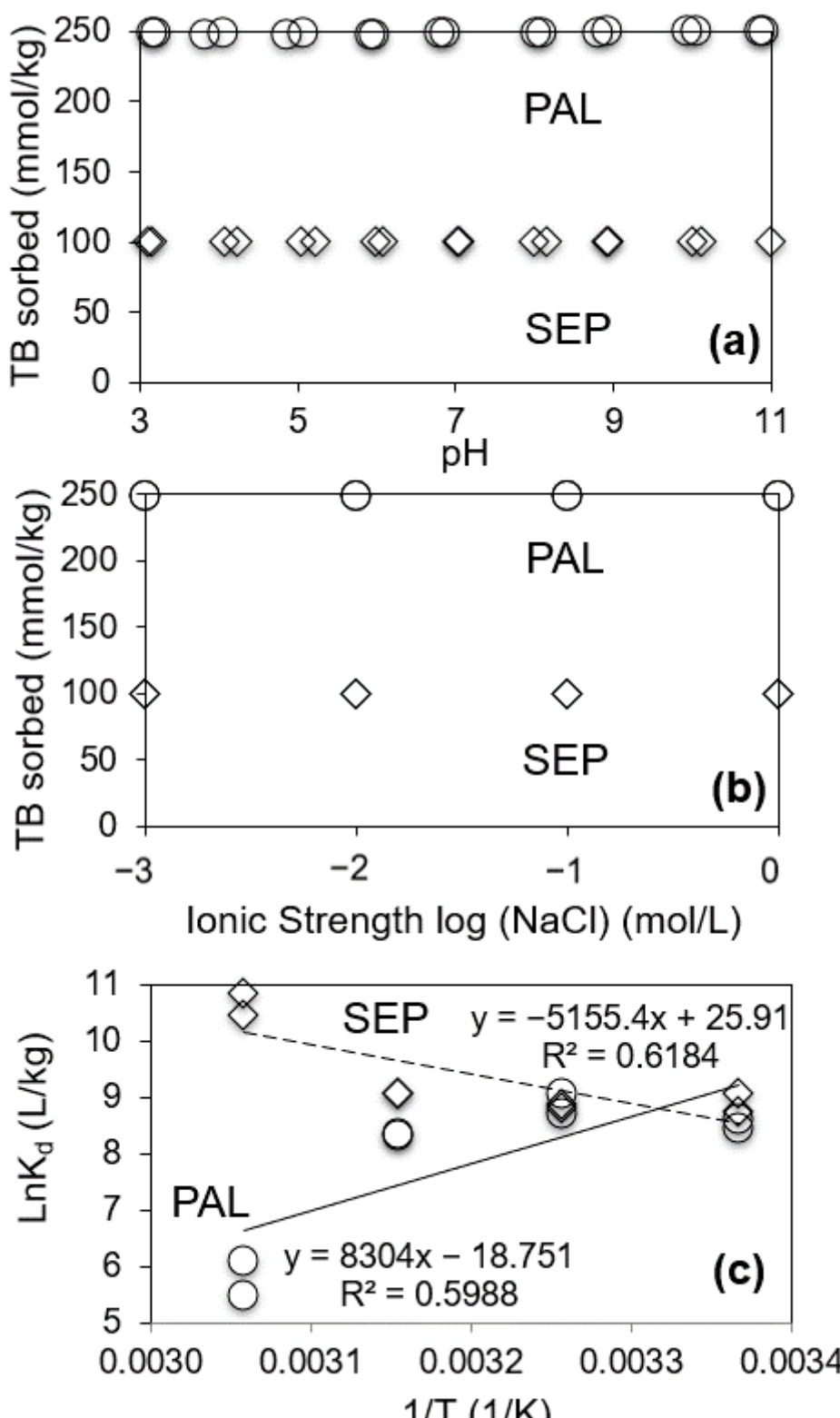

**Figure 4.** Influence of equilibrium solution pH (**a**), ionic strength (**b**), and temperature (**c**) on TB sorption on PAL and SEP.

**Table 1.** Thermodynamics of TB sorption on PAL and SEP.

| Minerals | $\Delta G$(kJ/mol) | | | | $\Delta H$(kJ/mol) | $\Delta S$ (kJ/mol-K) |
|---|---|---|---|---|---|---|
| | 296 K | 306 K | 316 K | 326 K | | |
| SEP | −20.9 | −23.1 | −25.2 | −27.4 | 42.9 | 0.2 |
| PAL | −22.6 | −21.0 | −19.5 | −17.9 | −69.0 | −0.2 |

### 3.4. XRD Analyses

The XRD patterns of PAL and SEP before and after TB sorption from different initial concentrations showed no changes in d-spacing (Figure 5), indicating the location of sorbed TB molecules was limited to the external surfaces, instead of intercalation, although both minerals have channel structures where the sites for the uptake of inorganic cations are available. Moreover, the peaks of crystalline TB were not observed in the XRD patterns of PAL and SEP, suggesting no TB precipitation into solid phase and the removal of TB from solution was due to sorption instead of crystallization. In comparison, the sorbed TB formed a single layer complex intercalated into the interlayer of MMT [24].

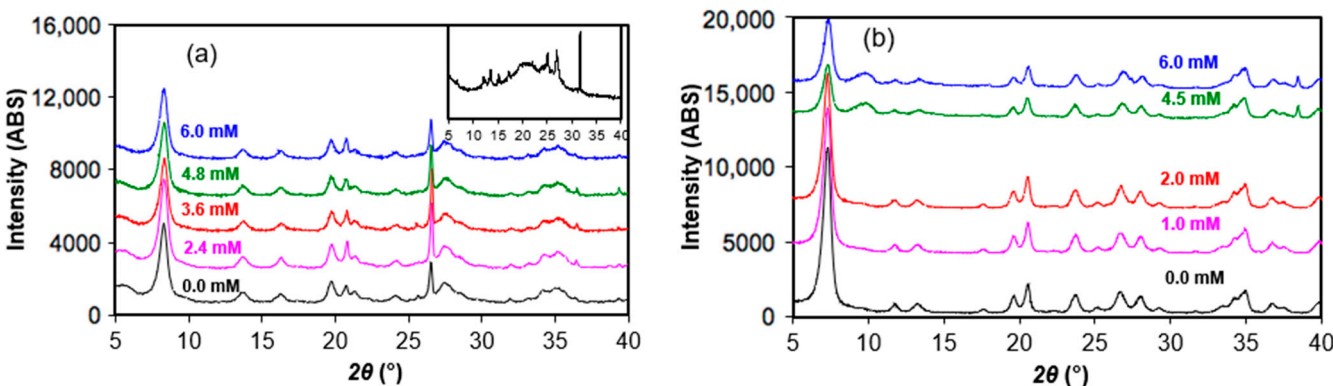

**Figure 5.** XRD patterns of PAL (**a**) and SEP (**b**) after TB sorption from different initial concentrations. The inset in is the XRD pattern of solid TB.

### 3.5. FTIR Analyses

The full FTIR spectra of both minerals and the solid dye TB are displayed in Figure 6. The band assignment for TB was lacking in the literature, although a full FTIR spectrum of TB is available on spectrabase.com (Wiley Spectrabase) (Hoboken, NJ, USA). For solid TB, the strong bands were located at 1597, 1387, 1319, and 885 cm$^{-1}$ (Figure 6). In addition, bands at 1474, 1418, 1225, and 1121cm$^{-1}$ were also medium to strong. The band at 1597 cm$^{-1}$ was assigned to the aromatic ring, and it shifted to 1603 cm$^{-1}$ after being doped on SiO$_2$ [25]. Similarly, the bands at 1601, 1492, and 1183 cm$^{-1}$ were assigned to the vibrations of the heterocycle skeleton of MB [26]. In this study, these bands were located at 1597, 1474, and 1225 cm$^{-1}$. The bands at 1597, 1387, 1319, and 1225 cm$^{-1}$ shifted to 1605, 1391, 1331 and 1234 cm$^{-1}$; and to 1607, 1387, 1333, and 1231 cm$^{-1}$ after TB sorption on PAL and SEP at high loading levels, as indicated by the small arrows pointing to the left below the FTIR spectrum of crystalline TB (Figure 7). The band at 1320 cm$^{-1}$ was attributed to C=N stretching in SO [27]. Thus, the shift of the band from 1319 to 1335 cm$^{-1}$ could be attributed to strong interactions between the C=N and the mineral surfaces, as a change in C=N stretching pointed to a perturbation of its environment [28]. Moreover, the band at 1325 cm$^{-1}$ was assigned to N-(CH$_3$)$_2$ [29]. The shift from 1319 to1335 cm$^{-1}$ suggested electrostatic interactions between N$^+$-(CH$_3$)$_2$ and the negatively charged mineral surfaces.

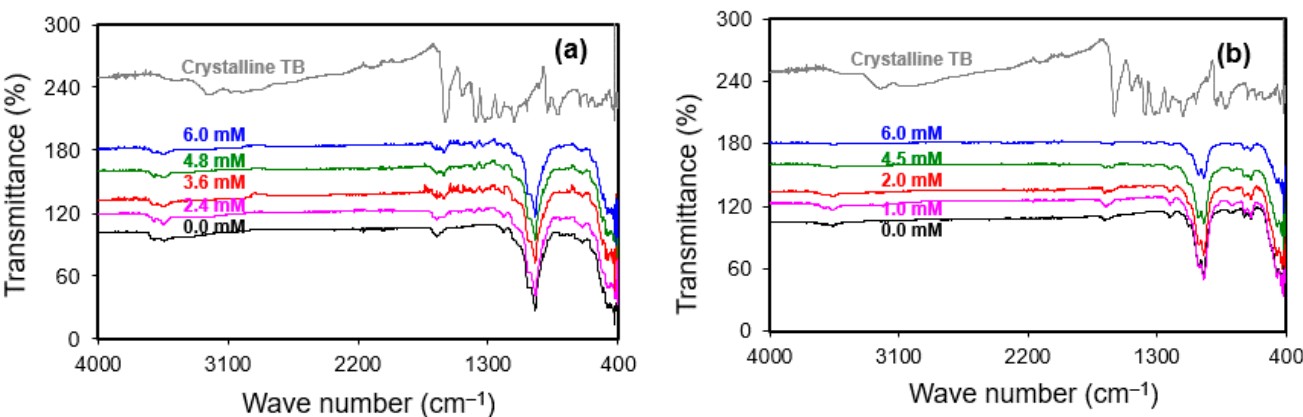

**Figure 6.** FTIR spectra of PAL (**a**) and SEP (**b**) after TB sorption from different initial concentrations.

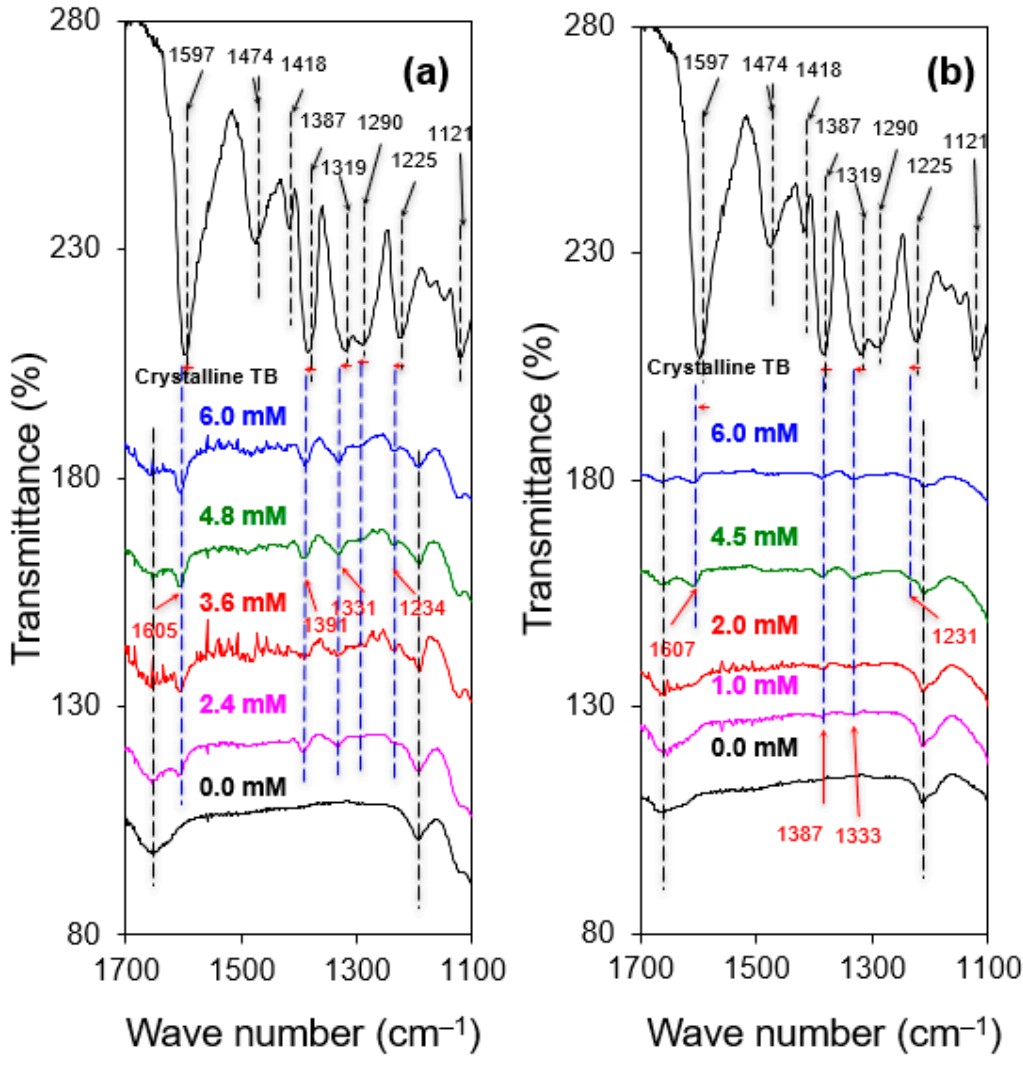

**Figure 7.** Enlargement of FTIR spectra in the wave numbers of 1100–1700 cm$^{-1}$ after TB sorption from different initial concentrations on PAL (**a**) and SEP (**b**).

### 3.6. Molecular Dynamic Simulation

To investigate the interactions between TB and mineral surfaces and to decipher the surface configuration of sorbed TB on minerals, molecular dynamic simulation was performed. The unit cell parameters are $a = 12.78$ Å, $b = 17.86$ Å, $c = 5.24$ Å, $\beta = 95.78°$,

Z = 4; and *a* = 13.43 Å, *b* = 26.88Å, *c* = 5.281 Å, Z = 4 for PAL and SEP, respectively. To ensure enough numbers of dye molecules in the simulation while not slowing down the simulation calculation, a supercell made of $2b \times 4c$ was used for the simulation. The XRD results showed no changes in d-spacing after TB sorption on both minerals, indicating that the sorption sites were limited to the external surfaces. Thus, at the TB sorption capacity, the SSA values of 173 [22] and 250 m$^2$/g [13] were used for PFL and SEP to determine the numbers of TB molecules used per supercell and the values were 7 and 3 per supercell for PAL and SEP, respectively. The results showed that the sorbed TB molecules formed a closely packed monolayer configuration on PAL (Figure 8). If the SSA value of 136 m$^2$/g were used, the number of TB molecules per supersell would be about 10, and the TB configurations on PFL would be more compact and some dimer formations of TB might form on PFL surfaces. In contrast, at the TB sorption maximum, isolated monomers or dimers formed on SEP surfaces due to its much lower CEC and large SSA values (Figure 9). The dimeric formation from the simulation agreed well with the Cl$^-$ sorption accompanying TB sorption as evidenced by the equilibration counterion Cl$^-$ concentration analyses (Figure 3). The dimension of TB molecule is 1.2 nm by 0.52 nm by 0.12 nm. Thus, at a flat lying configuration, the space occupied per TB molecule would be 0.6 nm$^2$, while at the TB sorption capacity, the space available for TB is about 1 nm$^2$ per molecule for PAL. In contrast, the available space of SEP for TB sorption is about 3.5 nm$^2$ per molecule. As such, loosely packed monomers were anticipated and confirmed in the molecular dynamic simulations. Moreover, dimer sorption on SEP is also confirmed by the simulation results (Figure 9). Another feature obtained from the simulation shows that the elongation of the molecule is parallel to the surface. However, molecular dynamic simulation showed that for most of the time, the flat surface of the molecule was not parallel to, but inclined towards the surface of PAL (Video S1 in Supplementary Materials). This inclination might be attributed to the electrostatic interaction between the positively charged dimethyl ammonium and the negatively charged mineral surfaces. For SEP, the surface is large enough to accommodate flat-lying TB molecules. However, the charge density of SEP is much lower than that of PAL. As such, the interaction between TB and the SEP surface would be much weaker. TB dimer formation or self-dimeric association is anticipated and confirmed by the molecular dynamic simulation (Video S2 in Supplementary Materials). This dimeric association is responsible for the additional TB uptake beyond the CEC value of SEP and for the large uptake of Cl$^-$ accompanying TB sorption.

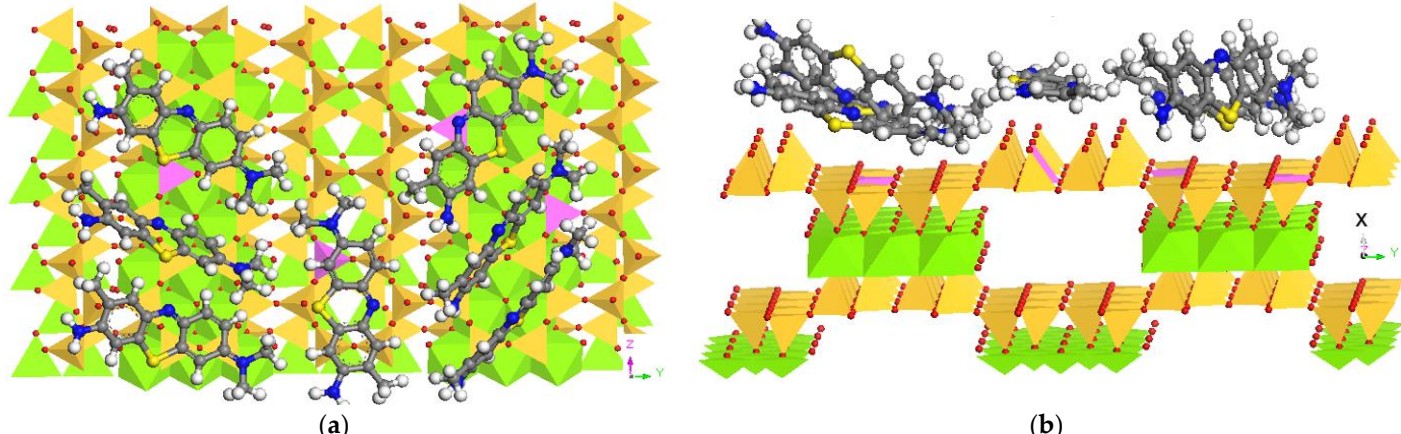

**Figure 8.** Molecular dynamic simulation showing the surface configuration of TB aggregates on the {100} surface (**a**) and projection along [001] (**b**) of PAL with 4 Si replaced with 4 Al in the supercell.

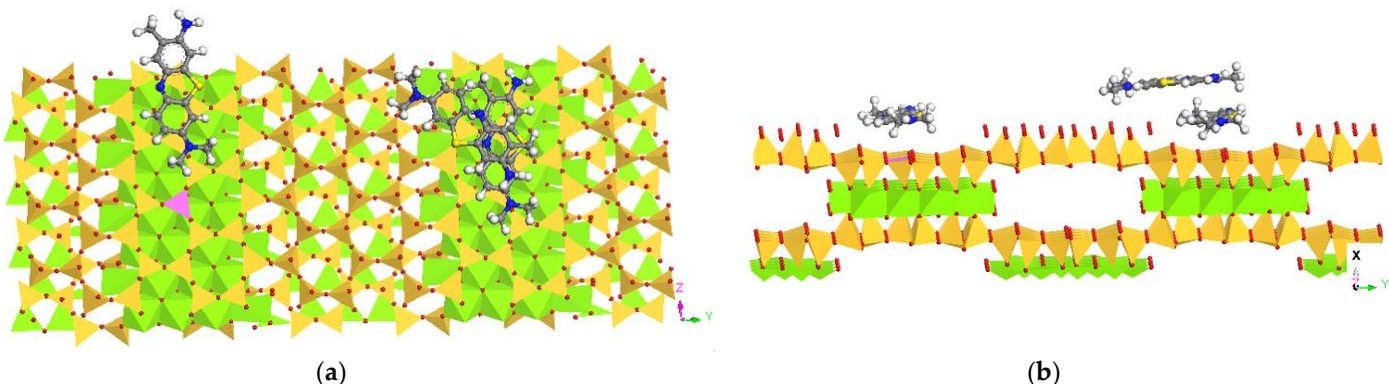

(**a**)                                    (**b**)

**Figure 9.** Molecular dynamic simulation showing the surface configuration of TB aggregates on the {100} surface (**a**) and projection along [001] (**b**) of SEP with 1 Si replaced with 1 Al in the supercell.

### 3.7. Discussions

The TB sorption capacities on PAL and SEP were 287 and 120 mmol/kg (Figure 2a). Meanwhile, their CEC values are 175 [10] and 15 [13] meq/kg for PAL and SEP, respectively. The TB is a monovalent cation $TB^+$ in the tested pH range (Figure 1b). MB has a very similar molecular structure to TB. The sorption of MB on PAL and SEP also far exceeded their CEC values with MB sorption as high as 350 mmol/kg on PAL (the same PFl-1 palygorskite used in this study) and up to 400 mmol/kg on SEP (from Turkey with a CEC of 110 meq/kg and SSA of 384 $m^2$/g) [30]. Similarly, in a different study, the sorption of MB on SEP could be up to four times its CEC value [31]. Thus, cation exchange alone cannot explain the much larger TB sorption capacities in comparison to their CEC values. The results strongly suggest that beyond cation exchange, other mechanisms must play a crucial role in TB sorption. The additional MB sorption on PAL and SEP was attributed to the neutral sorption site, whose contribution was about the same magnitude as that from the charged site [30,31]. On the contrary, the MB was studied for the determination of the CEC values of MMT with reasonable results if the initial MB concentration was less than 4 mM and the total input was not more than 150% of the CEC of MMT [32]. Sorption of TB decreased the CEC value of MMT significantly [24]. Moreover, no changes in *d*-spacings at the TB sorption capacities as confirmed by the XRD results (Figure 5) suggest surface sorption of TB on both minerals. The SSA values are 173 [22] and 250 $m^2$/g [13] for PAL and SEP, respectively. The dimension of TB molecules is about 1.2 nm long by 0.51 nm wide by 0.12 nm thick (Figure 1). Using these values and the TB sorption capacities, the area occupied per TB molecule would be about 1 and 3.5 $nm^2$. This would indicate that the SSA values of the minerals are large enough to accommodate a flat lying monolayer coverage. The molecular dynamic simulation showed close packing of TB molecules on the {100} surface of PAL (Figure 8). This close monolayer packing may also be partially attributed to the dimers of TB on mineral surfaces, as indicated by the parallel arrangement of two TB molecules on the surface (Figure 8) and motions in animation (Movie S1 in Supplementary Materials) based on a molecular dynamic simulation. In contrast, loose TB packing was observed on SEP due to its large SSA values (Figure 9). However, the simulation showed that the TB monomers tried to associate themselves with dimers to minimize the interaction energy (Movie S2 in Supplementary Materials). As such, sorption of TB dimers played a significant role in the higher TB sorption in comparison to the CEC values of the minerals.

The reported dimerization constant $K_D$ is 3311 $M^{-1}$ [33], which is defined as:

$$K_D = \frac{[D]}{[M]^2} \tag{7}$$

where $[D]$ and $[M]$ are the concentrations of dimers and monomers, respectively. At the highest input TB concentration of 6 mM, the calculated $[M]$ and $[D]$ are 0.88 and 2.56 mM. The preferred configuration of TB dimeric aggregation is the H type in an antiparallel

fashion [34]. The $K_D$ value for MB is 2380 M$^{-1}$ [35]. With this value, if the initial MB concentration was 6 mM, the [M] and [D] would be 1.02 and 2.49 mM. As such, the additional sorption of MB might also be attributed to dimer formation on the mineral surfaces, which the authors attributed to cation sorption to neutral sites [30].

## 4. Conclusions

The interactions between a cationic dye toluidine blue and fibrous clay minerals palygorskite and sepiolite in aqueous solution were studied under different experimental conditions. The isotherm study showed that the TB sorption capacity was much higher than the CEC values of the minerals, particularly for SEP. As such, cation exchange alone cannot explain the much larger TB uptake beyond the CEC values. The SSA values of the minerals played a certain role in the uptake of TB on both minerals. Most importantly, it is the dimeric formation of TB on both minerals that contributes to the significantly higher TB uptake. Other physico-chemical parameters such as equilibrium solution pH and ionic strength had almost no effect on TB sorption. The invariable d-spacing after TB sorption on both minerals suggests that the sorption sites were limited to the external surfaces of the minerals. The FTIR results suggested participation of N$^+$ in the electrostatic interaction with the negatively charged mineral surface. Overall, the PAL is a good candidate for sorptive removal of cationic dye from water in comparison to SEP.

**Supplementary Materials:** The following are available online at https://www.mdpi.com/article/10.3390/cryst11060708/s1, Video S1: Animation of TB sorption on PAL. Video S2: Animation of TB sorption on SEP.

**Author Contributions:** Conceptualization, Z.L. and Q.W.; methodology, Z.L.; software, X.W.; validation, Z.L., Q.W. and X.W.; formal analysis, K.C.; investigation, Q.C.; resources, Z.L.; data curation, Z.L., K.C. and Q.C.; writing—original draft preparation, Z.L.; writing—review and editing, Q.W.; visualization, Z.L. and X.W.; supervision, Z.L.; project administration, Z.L.; funding acquisition, Z.L. and Q.W. All authors have read and agreed to the published version of the manuscript.

**Funding:** The research was partially supported by the National Natural Science Foundation of China (Grant 41403083), the Key Project of Scienceand Technology of Hubei Provincial Department of Education (Grant D20141305), the Science Foundation of Education Commission of Hubei Province of China (Grant T2020008) and by a SPARK grant provided from Wisys.

**Institutional Review Board Statement:** Not applicable.

**Informed Consent Statement:** Not applicable.

**Data Availability Statement:** Upon request.

**Conflicts of Interest:** The authors declare no conflict of interest.

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
