# Peer review of "Interactions between Cationic Dye Toluidine Blue and Fibrous Clay Minerals"

_crystals, doi:10.3390/cryst11060708_

Round 1
Reviewer 1 Report
The manuscript present results on the adsorption of a cationic dye onto fibrous clay minerals. The paper contains interesting results and a wide enough experimental work, however, the discussion of experimental data should be enlarged and refined before publication.
General comments:
Key properties (surface area and CEC) of the used clays are reported based on a mix of literature sources and suppliers website.
Even if the supplier Source Clay Minerals Repository claims an effort to homogeneize its stock of standard minerals (which includes palygorskite), some variability is to be expected. Moreover, the surface area reported on their website is 136 mq/g, different from the data reported in the manuscript (https://www.clays.org/sourceclays_data/).
As for sepiolite, it is included by the supplier in the list of special clays, with low production volume and no homogeneization attempt (https://www.clays.org/source-clays/). Surface area and CEC are not reported for these materials on the website.
Are the authors confident in the data provided, not directly measured on their stock of clay but derived from papers published >15 years ago? Did they experimentally check the values?
The adsorption results where not dependent on pH or ionic strength and showed an opposite dependence on temperature (thus suggesting a different adsorption thermodynamics) for PAL and SEP. These points should be better discussed, as:
- a pH dependence has been observed in similar systems (e.g. Ref. 4 and 11 in the manuscript)
- both clays have a relatively similar surface and structure, so a completely different adsorption behaviour is not expected and should be further investigated/commented
- the absorption shifts shown in FTIR spectra are the same in both systems, suggesting similar interactions are established
- a dimerization mechanism is suggested, mediated by Cl- ions. Why this mechanism should be more active in the SEP systems? Shouldn't it be sensitive to pH?
The authors are invited to deepen the discussion on these points into the manuscript, clarifying the discussion of results.
Reviewer 2 Report
The manuscript is potentially intersting, but before publication the authors must address a few points. 1. error bars in the reported data are needed, or at least an estimate of teh uncertainties. 2.the discussion f IR spectra is confusing. Extracting precise frequencies from the spectra is difficult, mainly in view of the large, partially overlapping bands in the crystalline TB. I wonder if Raman spectra could be more informative. In any case, I do not understand how an hardening of a C=N stretching moe can be ascribed to increased interactions between teh molecule and teh surface. 3. MD simulation details are needed: is teh calculation done in water? In case which is teh total water molecule? in which conditions were the simulations run? how long?Author Response
Please see the attachment.

Round 2
Reviewer 1 Report
The authors have addressed the questions presented, adding details to the manuscript that help supporting their discussion. The paper can be now considered for publication
Reviewer 2 Report
Figures are missing in the revised ms and equations do not appear well. That said I do not understand how an MD calculation run without water can tell anything about the system. Overall the presented data are not convincingy supporting the conclusions.